# Metabolite trafficking enables membrane-impermeable-terpene secretion by yeast

So-Hee Son[1,2,7], Jae-Eung Kim[1,7], Gyuri Park[3], Young-Joon Ko[1], Bong Hyun Sung[4], Jongcheol Seo[5], Seung Soo Oh [2,3,6✉] & Ju Young Lee [1✉]

Metabolites are often unable to permeate cell membranes and are thus accumulated inside cells. We investigate whether engineered microbes can exclusively secrete intracellular metabolites because sustainable metabolite secretion holds a great potential for mass-production of high-value chemicals in an efficient and continuous manner. In this study, we demonstrate a synthetic pathway for a metabolite trafficking system that enables lipophilic terpene secretion by yeast cells. When metabolite-binding proteins are tagged with signal peptides, metabolite trafficking is highly achievable; loaded metabolites can be precisely delivered to a desired location within or outside the cell. As a proof of concept, we systematically couple a terpene-binding protein with an export signal peptide and subsequently demonstrate efficient, yet selective terpene secretion by yeast (~225 mg/L for squalene and ~1.6 mg/L for β-carotene). Other carrier proteins can also be readily fused with desired signal peptides, thereby tailoring different metabolite trafficking pathways in different microbes. To the best of our knowledge, this is the most efficient cognate pathway for metabolite secretion by microorganisms.

[1] Research Center for Bio-based Chemistry, Korea Research Institute of Chemical Technology (KRICT), 406-30, Jongga-ro, Jung-gu, Ulsan 44429, Republic of Korea. [2] School of Interdisciplinary Bioscience and Bioengineering, Pohang University of Science and Technology (POSTECH), Pohang, Gyeongbuk 37673, Republic of Korea. [3] Department of Materials Science and Engineering, Pohang University of Science and Technology (POSTECH), Pohang, Gyeongbuk 37673, Republic of Korea. [4] Synthetic Biology and Bioengineering Research Center, Korea Research Institute of Bioscience and Biotechnology (KRIBB), Daejeon 34141, Republic of Korea. [5] Department of Chemistry, Pohang University of Science and Technology (POSTECH), Pohang, Gyeongbuk 37673, Republic of Korea. [6] Institute for Convergence Research and Education in Advanced Technology (I-CREATE), Yonsei University, Incheon 21983, Republic of Korea. [7] These authors contributed equally: So-Hee Son, Jae-Eung Kim. ✉email: seungsoo@postech.ac.kr; juylee@krict.re.kr

Microorganisms exocytose certain metabolites for maintaining homeostasis as well as for regulating overflow metabolism[1]. When metabolic pathways are highly imbalanced, intracellular metabolites can diffuse, sometimes assisted by membrane transport proteins, across membranes into an extracellular medium. Secretion of toxic metabolic products offers an effective way for host cells to relieve the feedback inhibition of metabolic pathways by replenishing intracellular reservoirs[2–4]. This indicates that regardless of toxicity, value-added metabolites can be consistently produced and transported out of genetically modified microbial cells[5]. The desired compounds can then be easily recovered from an extracellular liquid medium, without any harvesting or cell disruption. Thus, metabolite secretion engineering holds great potential for many applications in synthetic biology, including continuous flow production of valuable metabolites in an efficient and cost-effective manner[6,7].

Despite industrial and pharmaceutical importance, microbial engineering for selective metabolite secretion is exceptionally challenging. Recently, the use of ATP-binding cassette transporters, known to export lipids or drugs, was explored to mediate biofuel efflux in Escherichia coli[8–11]. However, exocytosis of isoprenoids by polyspecific noncognate transmembrane proteins was indiscriminate, and the degree of efflux was insignificant (~25 mg/L)[12]. In this regard, interorganellar protein trafficking in eukaryotes is unique[7,13,14]. Thousands of cytoplasmic proteins are encrypted with sorting signals, and due to these address labels, each protein can accurately be delivered to intracellular compartments or extracellular milieu. This is the most significant systemic difference between the secretion of proteins and metabolites.

In this study, we investigated whether metabolite secretion could be selectively guided by discrete sorting signals, similar to those of the elaborate protein trafficking system. It is well known that all metabolites interact with proteins as they serve as either substrates or products in enzymatic reactions, and allosteric proteins require specific ligands. Molecular recognition can appropriately be exploited for metabolite trafficking; specific proteins that bind to desired metabolites can be chosen and readily tagged with signal peptides for further sorting. Thus, the fusion proteins, capable of carrying the target metabolites, are designated certain destinations, within or outside cells. By integrating a metabolite-binding protein with its sorting tag, we successfully invented a metabolite trafficking system. As a proof of concept, we systematically tailored an exclusive pathway for the secretion of medicinal terpenes, large and hydrophobic high-value chemicals, and demonstrated that terpene secretion was substantially boosted in Saccharomyces cerevisiae (e.g., ~226 mg/L for 6-day batch fermentation and ~670 mg/L for 15-day semi-continuous culture). To the best of our knowledge, this is the most efficient cognate pathway for selective metabolite secretion in microorganisms, thus enabling the intracellularly-accumulated target compounds to pass through otherwise impermeable membranes.

## Results

### Tailoring a metabolite trafficking pathway for selective terpene secretion

Our fusion protein design involved the coupling of a terpene-binding protein with an export signal peptide. Since supernatant protein factor (SPF) is involved in the regulation of cholesterol biosynthesis in the human liver[15], we identified this cytosolic lipid-binding protein as a terpene carrier. To facilitate the hydrophobic interaction of SPF with lipophilic metabolites, we only employed the lipid-binding domain of SPF (tSPF, amino acids 1–278) by eliminating the C-terminal Golgi dynamics

domain (Supplementary Fig. 1)[16]. In parallel, we examined the ability of sucrose transport protein (invertase, Suc2), derived from yeast, for export signaling via co-translational translocation[17]. When the signal peptide of Suc2 is cleaved from a nascent protein in the endoplasmic reticulum (ER) lumen by signal peptidases, the remaining secretory protein is immediately translocated into the ER membrane for subsequent secretion. Taking advantage of this mechanism, we fused the Suc2 signal peptide to the tSPF N-terminal to obtain Suc2-tSPF. This tailored fusion protein could perform a series of coordinated actions for selective terpene secretion (Fig. 1 and Supplementary Fig. 2). Briefly, once Suc2-tSPF was translated in the ER (step a), signal peptidases subsequently cleaved the Suc2 signal peptide (step b). The mature tSPF was loaded with terpenes (step c) and then transported to the Golgi (step d). To complete terpene trafficking, the terpene-loaded tSPF was exported into the extracellular medium (step e).

### Extracellular secretion of membrane-impermeable terpenes by yeast

We successfully expressed Suc2-tSPF in an engineered S. cerevisiae strain for squalene overproduction and achieved extracellular secretion of hitherto membrane-impermeable terpenes (Fig. 2). Briefly, the Suc2-tSPF gene was systematically incorporated into our original squalene producer (SQ03-INO2; subsequently referred to as SQ), capable of producing up to 69 mg/g dry cell weight (DCW) squalene in 144 h, which was nearly three orders of magnitude higher than that in the wild-type strain[18]. In yeast, squalene is synthesized in the ER as the enzymes relevant to squalene biosynthesis are mainly ER-located[19]. Therefore, when both cargo (squalene) and carrier (tSPF) are synthesized and bound to each other in the ER, the export signal (Suc2 signal peptide) readily guides the squalene-loaded tSPF protein for the actual export across the cell membrane impermeable to the squalene. As all the events occur in the same location, i.e., the ER, it ensures effective loading of the squalene products into the Suc2-tSPF export carriers and subsequent secretion of the squalene-loaded tSPF proteins to extracellular spaces.

Unlike conventional methods, a two-phase cultivation system comprising a culture medium and 10% (v/v) immiscible dodecane was adopted (Fig. 2A)[12,20]. Hydrophobic metabolites are soluble in the organic phase, but not in the aqueous medium; thus, the secreted squalene can be fully recovered by simply collecting the dodecane (see Methods and Supplementary Note 1). High-performance liquid chromatography (HPLC) revealed that successful overexpression of Suc2-tSPF resulted in remarkably high levels of squalene in the dodecane phase (Fig. 2B (navy) and Supplementary Fig. 3). In contrast, no squalene secretion was observed when the control strain SQ was used (blue).

Surprisingly, the significantly improved secretion of squalene was demonstrated by the newly engineered yeast strain (Suc2-tSPF-expressing SQ and Suc2-tSPF/SQ). Time-dependent quantification of squalene secretion (Fig. 2C, Supplementary Fig. 4 and Supplementary Table 1) revealed a significant increase in extracellular squalene secretion by Suc2-tSPF/SQ compared with the original SQ. After 144 h of cultivation, the titer of extracellular squalene produced by Suc2-tSPF/SQ was 226 mg/L, which was ~26-fold more than that by SQ (8 mg/L). We noted that long-term cultivation (6 days) was accompanied by the natural death of old cells, which were inevitably included in the dodecane phase. Thus, SQ yielded markedly low levels of extracellular squalene via inadequate squalene secretion. Suc2-tSPF/SQ also exhibited increased intracellular squalene accumulation (Supplementary Table 1); it produced 97 mg/g DCW of intracellular squalene after 144 h of fermentation, and this value showed a ~40% increase over that produced by the original SQ (69 mg/g

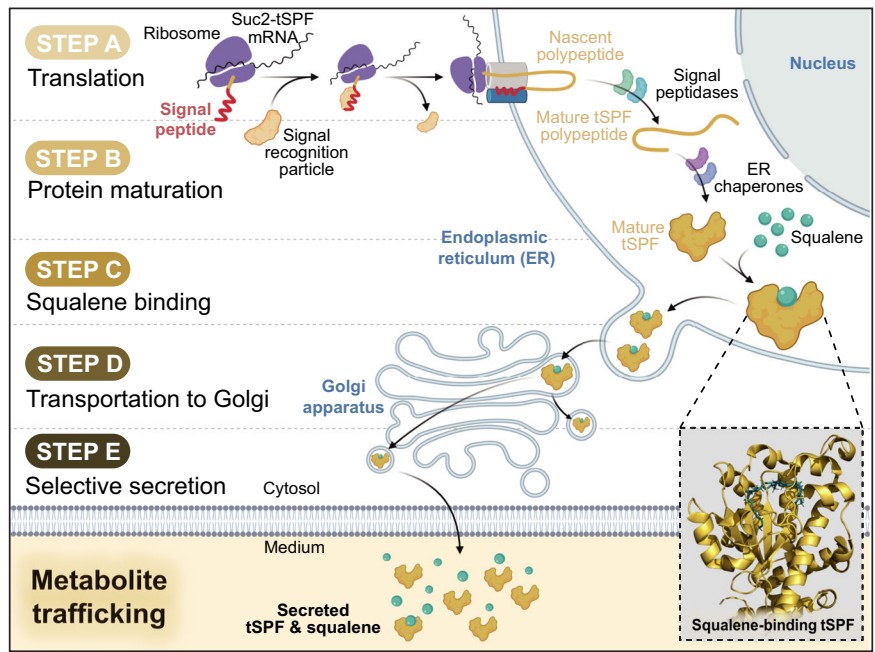

**Fig. 1 Metabolite trafficking pathway for selective terpene secretion.** By combining a squalene-binding protein, a lipid-binding domain of supernatant protein factor (tSPF), and the export signal peptide of sucrose transport protein (Suc2), we systematically designed a fusion protein, Suc2-tSPF. Lipophilic squalene could be loaded into the Suc2-tSPF and transported into the extracellular milieu, across the otherwise impermeable membrane. To achieve selective terpene secretion, Suc2-tSPF performs a series of coordinated actions. During Suc2-tSPF mRNA translation, the N-terminal signal peptide is bound to a signal recognition particle for co-translational translocation, and the complex of the ribosome with the partially translated Suc2-tSPF is transferred to the endoplasmic reticulum (ER) membrane (**step a**). A nascent polypeptide is synthesized, and the signal peptide is cleaved by signal peptidases in the ER lumen, leaving behind a tSPF polypeptide (**step b**). The tSPF polypeptide is further modified by ER chaperones to form a mature tSPF protein, which captures squalene, the target metabolite (**step c**). Once squalene succeeds in hitchhiking, the tSPF protein, encapsulated by a transport vesicle, is subsequently transported to a Golgi apparatus (**step d**). To finalize squalene trafficking, a secretory vesicle exports the squalene-carrying tSPF into the extracellular space (**step e**).

DCW). Without the signal peptide, tSPF alone could not secrete squalene (Fig. 2B, C and Supplementary Table 1). The tSPF-expressing SQ strain (tSPF/SQ) showed negligible squalene secretion (13 mg/L), similar to that of the control SQ strain (8 mg/L), even after fermentation for 144 h.

**Signal peptide-guided exclusive secretion of intracellularly-accumulated terpenes.** Selective metabolite secretion depends on the level of each intracellular product, so highly accumulated squalene was exclusively secreted (Supplementary Note 2). Unlike squalene, 2,3-oxidosqualene, an important metabolic intermediate in the squalene biosynthesis pathway, was not secreted. Considering that SPF has no significant preference for either squalene or 2,3-oxidosqualene, we speculated that their intracellular accumulation would determine whether they could be loaded onto the export carrier, Suc2-tSPF. Since 2,3-oxidosqualene was rapidly consumed in our engineered yeast cells, its intracellular concentration was not sufficient to trigger Suc2-tSPF capture for subsequent secretion (Supplementary Table 2). Conversely, the extracellular transport of the desired product, squalene, increased in a time-dependent manner. On comparing the results of Suc2-tSPF/SQ cultivation for 72 and 144 h, we found that extracellular squalene was increased by ~36%, confirming that squalene titers gradually increased over time (Fig. 2C and Supplementary Table 1).

Even when other export signal peptides were coupled with tSPF, the selective squalene secretory pathway remained active. Two different proteins, acid phosphatase (Pho5) and yeast α-mating factor (MFα) (Supplementary Table 3) were chosen on the basis of translocation modes[21–23]. Pho5 uses the same co-translational translocation pathway as Suc2, but its signal peptide is cleaved by different ER peptidases. In contrast, MFα is transported to the ER membrane via posttranslational translocation, which involves two-step cleavage of the signal peptide (Supplementary Fig. 2). Following the removal of the export signal peptide of MFα by signal peptidases in the ER lumen, Kex2, and Ste13 proteases further cleave the remnant of the signal peptide in the Golgi. Owing to the same translocation mode, squalene secretion by Pho5-tSPF was comparable to Suc2-tSPF (Fig. 2B); however, total squalene secretion after 144 h of cultivation (190 mg/L) was lower (Fig. 2C and Supplementary Table 1). Interestingly, MFα-tSPF, which is trafficked via a route distinct from that of Suc2-tSPF, showed significantly lower squalene secretion (Fig. 2B, C and Supplementary Table 1). Although the fundamental mechanism underlying this is unclear, we speculated that the inherent complexity of signal peptide cleavage might have affected the loading efficiency of squalene onto tSPF before secretion[24].

**Protein hitchhiking-driven continuous secretion of terpenes.** To clarify further whether the presence of squalene in the dodecane phase was caused by secretion or cell lysis-dependent release, we performed a western blot assay (Fig. 2D, see Methods) using actin as a loading control. In principle, with negligible cell lysis, cytosolic actin should not be detected in the extracellular medium, but it must be possible to detect tSPF by signal peptide-guided secretion. Based on the actin bands of the culture supernatant, we verified that the presence of tSPF in the extracellular space was not caused by cell lysis. Without signal peptides, tSPF was enriched only inside the cells (Fig. 2D, upper). However,

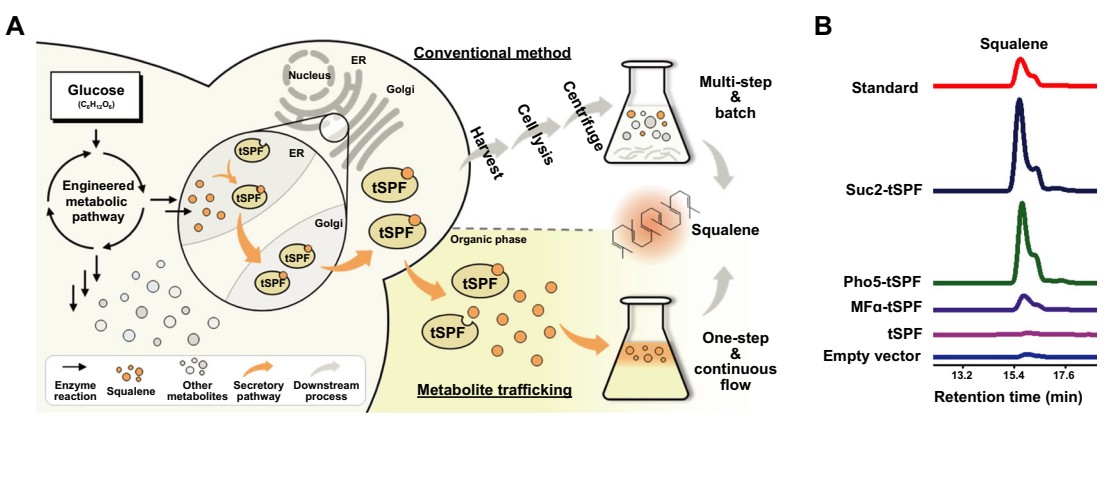

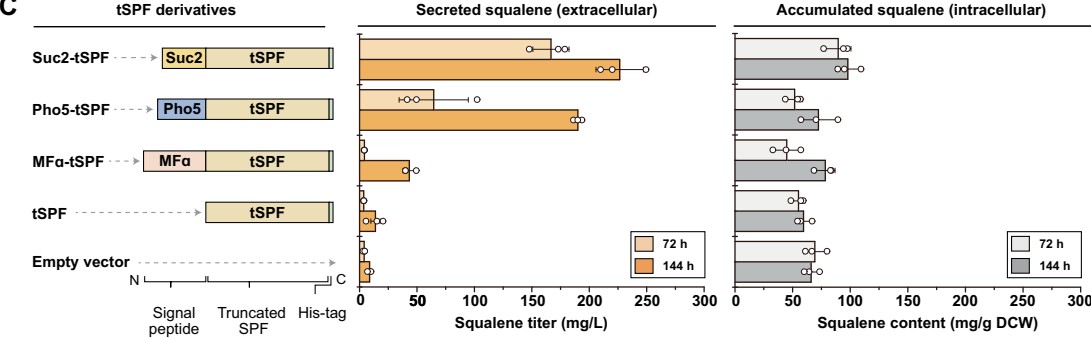

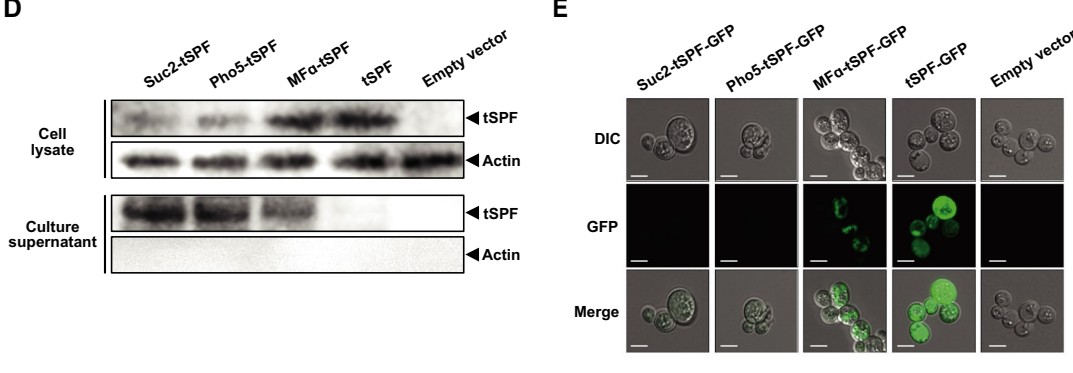

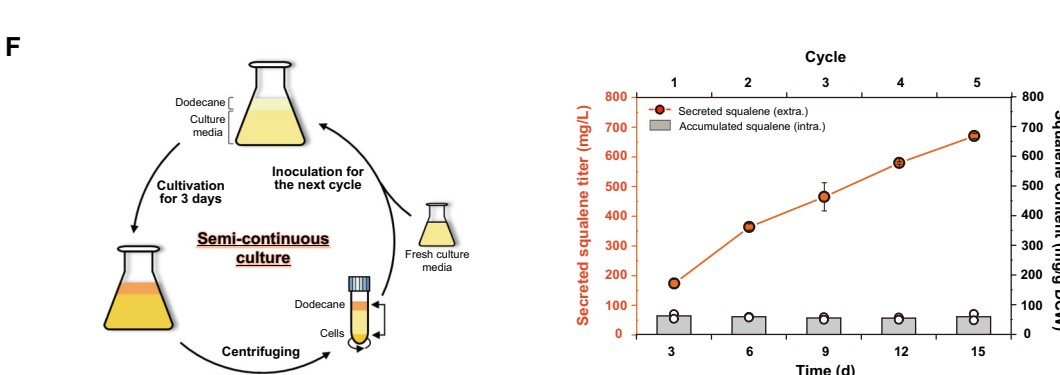

when the cells overexpressed tSPF that was systematically integrated with signal peptides, tSPF was successfully secreted out of the cells (Fig. 2D, lower). Among the three different signal peptides, Suc2 exported tSPF most efficiently, which was consistent

with the finding that Suc2-tSPF/SQ secreted squalene most productively.

We further validated the role of signal peptide-guided tSPF in squalene secretion by fusing green fluorescent protein (GFP) to the

**Fig. 2 Signal peptide-guided tSPFs selectively transport loaded squalene to the extracellular medium. A** Schematic illustration of squalene production and recovery by our metabolite trafficking strategy. An SQ03-INO2 yeast strain (SQ), capable of squalene overproduction, was engineered to overexpress tSPF derivatives tagged with export signal peptides (left). Unlike conventional methods, such as cell disruption, which rely on multiple steps for squalene extraction (right, top), the signal peptide-tagged tSPF derivatives enable continuous flow production of squalene (right bottom) in an energy-efficient and cost-effective manner. **B** High-performance liquid chromatograms of squalene secreted by tSPF derivatives. At the same retention time of the squalene standard (red), signal peptide-tagged tSPFs (Suc2-tSPF: navy; Pho5-tSPF: green; and MFα-tSPF: purple) yielded distinct peaks due to extracellular squalene secretion. When tSPF was neither tagged with signal peptides (pink) nor expressed in yeast (blue), only negligible peaks were detected. **C** Quantitative measurements of secreted and accumulated squalene. SQ strains included each export signal peptide and tSPF genes (left). Extracellular and intracellular squalene (middle and right) was quantified by collection of the dodecane phase and disruption of the harvested cells, respectively. **D** Western blot analysis of cell lysates (top) and culture supernatants (bottom). With signal peptides, most tSPFs migrated into the culture medium, but without them, the tSPF stayed in the cell. **E** Differential interference contrast (DIC) and confocal fluorescence microscopy images of yeast cells that overexpressed tSPF derivatives fused with a green fluorescent protein (GFP). Owing to extracellular secretion, fluorescent GFP-fused Suc2-tSPF and Pho5-tSPF were not accumulated inside the cells. In comparison, MFα-tSPF was not efficiently secreted from the cell; however, the non-tagged tSPF was completely reserved inside the cells. Scale bar: 5 μm. **F** Semi-continuous fermentation system using Suc2-tSPF/SQ. With a continuous nutrition supply, yeast cells with Suc2-tSPF enable sustainable squalene production and secretion. During semi-continuous culture, the growth medium was replenished every 3 days (left), and the levels of intra- and extracellular squalene were monitored for 15 days (right). All data represent the mean of biological triplicates, and error bars indicate the standard deviation. Source data are provided in the Source Data file.

C-terminal of tSPF and observed its localization using confocal fluorescence microscopy (Fig. 2E and Supplementary Fig. 5). Corroborating the western blot results, the intracellular GFP signal was only intense for the signal peptide-lacking tSPF, whereas no fluorescence inside the cells were observed for the signal peptide-tagged tSPF. From this observation, we concluded that GFP-fused Suc2-tSPF and Pho5-tSPF (Suc2-tSPF-GFP and Phot-tSPF-GFP, respectively) enabled the secretion of tSPF-GFP after cleavage of the signal peptide. We noted that MFα-tSPF-GFP could not be completely transported outside the cells owing to its poor secretion capability. However, without the export signaling tag, the tSPF-GFP was destined for intracellular retention.

Indeed, the squalene capture of tSPF was responsible for the squalene secretion, which was validated by the use of control proteins incapable of squalene binding. For this purpose, we prepared Suc2-tagged GFP (Suc2-GFP), since GFP can neither perform extracellular secretion nor capture terpene (Supplementary Table 1)[25]. Overexpression of Suc2-GFP in the SQ strain clearly showed its extracellular export (Supplementary Fig. 6). However, there was no evidence of squalene secretion, thereby indicating that the engineered squalene secretion pathway was the exclusive result of synchronizing two different molecular functions, namely, protein hitchhiking and metabolite trafficking.

Additionally, we implemented a semi-continuous fermentation using Suc2-tSPF/SQ, the best squalene-secreting strain, to prove that our metabolite trafficking system could be used to produce and secrete the target metabolite in a continuous and efficient manner (Fig. 2F, Supplementary Fig. 7 and Supplementary Table 4; see Methods). Briefly, we sampled yeast cells and collected dodecane, replenishing with a fresh culture medium once every 3 days. During five repeated cycles, intracellular squalene remained at ~60 mg/g DCW, indicating no further increase in accumulation within the cells. However, extracellular squalene accumulated consistently, and by the fifth cycle, we reached the titer of the secreted squalene up to ~670 mg/L, the highest titer of squalene secretion reported to date. Moreover, the graph for total squalene production over time was linear, thereby indicating the potential for continuous flow production.

**Reprogramming the metabolite trafficking system for different terpene secretion.** Given the binding promiscuity of SPF towards other hydrophobic terpenes[16], we further expanded our innovative concept of metabolite secretion by applying it to the production of other terpenes. Thus, β-carotene, a tetra-terpene, was chosen to demonstrate the engineered secretion owing to its simplistic colorimetric detection[12]. For the production of β-carotene in *S.*

*cerevisiae* (Fig. 3A), we used the p415-BC plasmid containing four different β-carotene biosynthetic genes: *tHMG1* (truncated 3-hydroxy-3-methylglutaryl-CoA reductase 1) from *S. cerevisiae*, and *crtE* (geranyl diphosphate synthase), *crtYB* (phytoene synthase), and *crtI* (carotene desaturase) from *Xanthophyllomyces dendrorhous*. As ascertained by squalene secretion, we overexpressed Suc2-tSPF using yeast cells containing p415-BC for β-carotene production. Similar to the biosynthesis of squalene, that of β-carotene relies on ER-localized enzymes by their transmembrane domains[12,26], indicating that production of β-carotene and its subsequent secretion by Suc2-tSPF can be well synchronized in the ER. Interestingly, we observed a ~23-fold increase (1.4 mg/L) in β-carotene secretion compared to that by the control, which was incapable of secreting tSPF (0.06 mg/L) (Fig. 3B–D, Supplementary Figs. 3, 8 and Supplementary Table 5). Moreover, concurrent accumulation of lycopene, a precursor in the β-carotene biosynthesis pathway, induced the β-carotene-producing cells to secrete both β-carotene and lycopene simultaneously, suggesting that the terpene secretion is readily controllable by regulating the intracellular concentrations of terpenes (Supplementary Note 2).

Finally, we predicted the metabolites that could benefit from our SPF-driven secretion, using molecular docking with AutoDock Vina in PyRx[27,28]. We scrutinized 43 different terpenes that microbial cells have produced as production hosts[29,30] and the predicted binding energy was calculated for each compound (Fig. 3E and Supplementary Table 6, see Methods). The molecular interaction energy between SPF and each terpene varied from −10.7 to − 6.1 kcal/mol. Squalene and 2,3-oxidosqualene interaction energies were −10.6 and −10.5 kcal/mol, respectively, confirming their comparable binding capabilities to the SPF. In contrast, glutamate and pyruvate, key intermediates of metabolic pathways in yeast, exhibited much lower affinities (−4.4 and −3.6 kcal/mol, respectively), suggesting that the SPF carrier protein favors terpene binding. Furthermore, we analyzed the pattern of SPF-terpene interactions after the docking simulation; the positions of all terpenes overlapped within the SPF binding pocket (Supplementary Fig. 9), although end groups of several tetra-terpenes (e.g., an isophorone group of zeaxanthin) protruded from the SPF protein surface. These simulations suggest that SPF could serve as a suitable vehicle for various terpenes. Thus, the signal peptide-guided SPF would enable precise delivery of target terpenes to the desired location, including the extracellular space, thereby actualizing the synthetic pathway of metabolite trafficking.

## Discussion

In summary, we demonstrated that a common multiplexed protein secretion pathway could mediate sustainable and efficient

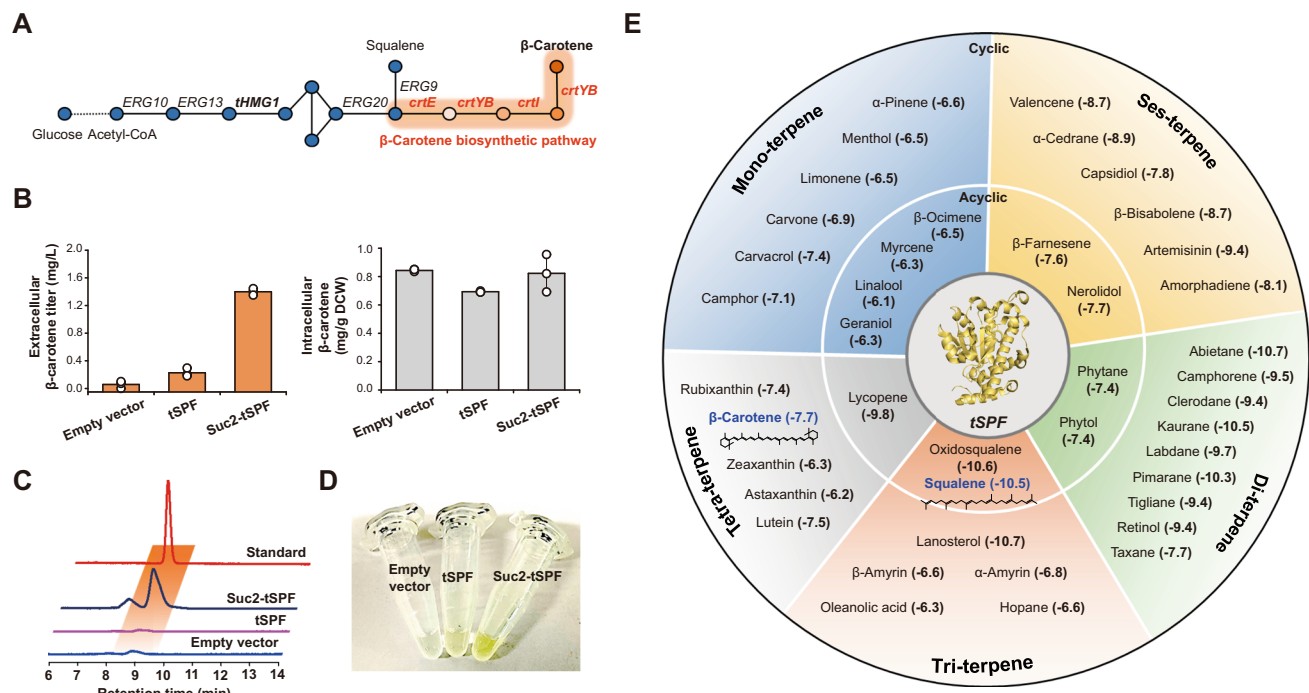

**Fig. 3 Extension of our metabolite trafficking strategy for the secretion of other terpenes. A** The biosynthetic pathway of β-carotene production. Compared to the squalene biosynthesis pathway, this pathway further involves the expression of three constitutive genes, namely *crtE*, *crtYB*, and *crtI*. **B** Quantitative measurements of secreted and accumulated β-carotene produced by the engineered strains that overexpress Suc2-tSPF. Yeast cells were grown in a defined minimal medium with 2% (w/v) glucose and 10% (v/v) dodecane at 30 °C for 144 h. All data represent the mean of biological triplicates, and error bars indicate the standard deviation. **C** High-performance liquid chromatograms of β-carotene secreted by Suc2-tSPF. Compared to the β-carotene standard (red), overexpression of Suc2-tSPF (navy) resulted in a distinct peak at the same retention time, thus validating the extracellular secretion of β-carotene. In contrast, non-tagged tSPF (pink) failed to transport β-carotene outside cells. **D** Colorimetric determination of secreted β-carotene, an orange-colored terpenoid. After 144 h of cultivation, the expression of Suc2-tSPF resulted in a visible color change of dodecane due to extracellular β-carotene secretion. However, when only tSPF or no carrier protein was expressed, the dodecane phase showed faint coloration. **E** The binding promiscuity of tSPF towards a wide range of hydrophobic terpenes produced by microbial cells. The binding energy between SPF and each terpene predicted by molecular docking with AutoDock Vina in PyRx varied from −10.7 to −6.1 kcal/mol, which was comparable to the binding energy between SPF and squalene. Source data are provided in the Source Data file.

extracellular transport of target metabolites. The integration of protein hitchhiking with metabolite trafficking was highly synergistic. Membrane-impermeable terpenes could be rescued from the extracellular medium without cell disruption and subsequent extraction, proposing a potentially cost-effective, high-yielding continuous flow process for the production of valuable chemicals, although more comprehensive optimizations of our secretion pathway are necessary for an industrial-scale application. Unlike previous noncognate transmembrane transporter engineering[12], our cognate secretion pathway achieved significantly improved terpene secretion (~226 mg/L for squalene and ~1.6 mg/L for β-carotene), the highest titer reported in microorganisms to date (Supplementary Note 3). In this study, the substantial improvement of terpene secretion may attribute to the high loading efficiency of the terpenes into tSPF carrier proteins (Supplementary Note 4), and further investigations of the in-depth secretion mechanism is ongoing.

For further improvement of terpene production and secretion, increasing the levels in overexpression of heterologous proteins relevant to terpene and carrier protein synthesis would be inevitably necessary, which is critical and not solved for our metabolite trafficking strategy. Moreover, as the underlying mechanisms of our carrier protein-driven terpene secretion has not been fully understood yet, the efforts to unravel the mechanisms would be also important. For the eventual applications, many different ways would be devised by systematically combining various synthetic biology tools and strategies (Supplementary Table 7 and Supplementary Note 5)[31–33]. For this proof of concept study, we overexpressed Suc2-

tSPF using the strongest yeast TDH3 promoter on a high-copy number plasmid, presumably causing the protein burden and the transport overload (Supplementary Note 6)[25,34]. This could be mitigated to some degree by balancing or tuning the expression levels of relevant proteins. Moreover, synthetic regulatory systems with diverse operational modes[35,36], such as auto-regulatory feedback loops, toggle switches, and engineered riboswitches, may permit the decoupling or integration of cellular growth and terpene secretion. Furthermore, as the expression level of carrier proteins is regarded as the main limiting factor that constrains the secretion level of terpenes, we envision that by further elevating tSPF levels, terpene secretion enhancement would be achievable (Supplementary Note 5).

Importantly, our metabolite trafficking strategy would be versatile by suggesting a conceptual methodology to secrete highly valuable, yet membrane-impermeable products to extracellular spaces. Despite the promiscuity of SPF, its concentration-dependent binding characteristic enables the desired metabolic product to be loaded only onto the carrier protein, thereby indicating that metabolic pathway engineering can allow us to readily choose the terpene to be secreted exclusively (Supplementary Note 2, 5). Furthermore, the use of other carrier proteins and signal peptides could be explored for our metabolite trafficking strategy with potential applications for metabolic engineering and synthetic biology.

## Methods

**Plasmid and strain construction**. All plasmids, strains, and primers used in this study are listed in Supplementary Tables 8, 9. The *tSPF* gene was codon-optimized

for *S. cerevisiae*, synthesized, and cloned into pD1214-FAKS (ATUM, formerly DNA 2.0; Newark, CA, USA), generating pSEC-MFα-tSPF. All other tSPF derivatives tagged with different export signal peptides in this study were generated with a set of primers replacing the MFα signal sequence with that of Suc2 or Pho5, resulting in pSEC-Suc2-tSPF and pSEC-Pho5-tSPF, respectively. For the expression of tSPF without any export signal peptides, the MFα signal sequence was deleted in pSEC-MFα-tSPF, generating pSEC-tSPF. We evaluated the signal peptide-tagged tSPF derivatives for selective squalene secretion by transforming corresponding tSPF-expressing plasmids into SQ strain, our squalene producer[18]. For β-carotene production in yeast, the plasmid pLM494, encoding β-carotene biosynthetic genes *crtE*, *crtI*, *crtYB*, and *tHMG1*, was obtained from Addgene (plasmid #100539). This pLM494 plasmid was digested with the restriction enzymes BamHI and SalI and then the resulting 11.3-kb fragment harboring expression cassettes for *crtE*, *crtI*, *crtYB*, and *tHMG1* was cloned into p415-GPD, yielding p415-BC. The p415-BC for β-carotene production and pSEC-tSPF or pSEC-Suc2-tSPF for β-carotene secretion were co-transformed into the wild-type CEN.PK2-1D. To visualize the locations of tSPF, pSEC-tSPF-GFP, pSEC-MFα-tSPF-GFP, pSEC-Pho5-tSPF-GFP and pSEC-Suc2-tSPF-GFP were constructed by insertion of a PCR fragment of tSPF, MFα-tSPF, Pho5-tSPF, or Suc2-tSPF, respectively, and the PCR fragment flanks SpeI and XhoI restriction enzyme sites into a p426-TEF1 vector, followed by the GFP gene insertion at the C-terminus of tSPF. The resulting plasmids were transformed into SQ. The standard LiAc/ssDNA/PEG method was used for yeast transformation[37].

**Terpene production by two-phase flask fermentation.** To examine terpene production and secretion by our metabolite trafficking, a two-phase culture system was adopted for flask fermentation of yeast cells. Yeast strains were grown on yeast synthetic complete (YSC) agar plates for 2–3 days at 30 °C and then transferred to 50 mL conical tubes with vent caps (SPL Life Sciences, Korea) containing 10 mL of YSC medium. YSC medium was composed of 0.67% (w/v) yeast nitrogen base without amino acids (BD Difco, USA), 0.19% (w/v) yeast synthetic drop-out medium supplements without uracil (Sigma-Aldrich, USA), and 2% (w/v) glucose. For β-carotene production, the drop-out medium supplement was substituted with the complete supplement mixture without leucine and uracil (MP Biomedicals, USA). Saturated overnight yeast cultures were used to inoculate 45 mL of YSC medium gently overlaid with 5 mL of dodecane (Sigma-Aldrich, USA) as an extractive solvent in 250 mL flasks to give an initial $OD_{600}$ of 0.5. For batch fermentation, two-phase cultures were incubated at 30 °C with shaking at 250 rpm for 6 days. For semi-continuous cultures, growing yeast cells and dodecane were collected and then transferred to a freshly prepared YSC medium every 72 h for 15 days. After cultivation, cells, culture media, and dodecane layers were collected for quantitative analysis of terpene production by HPLC. All flask fermentations were performed in three independent experiments.

**Terpene extraction from yeast cells and quantification.** Yeast cells were harvested by centrifugation at 13,000×*g* for 5 min. The harvested cells were resuspended in 0.6 mL of a 1:1 methanol-acetone solution. The mixture was transferred to a tube containing lysing matrix C, then disrupted mechanically using a FastPrep-24 5 G homogenizer (MP Biomedicals, USA) according to the manufacturer's instructions. For the quantification of terpenes, intracellular amounts were measured from the cell lysates, and extracellular (secreted) amounts were measured in the collected culture media or dodecane layers. Culture media and dodecane layers were prepared from cell cultures separated by centrifugation at 4000×*g* for 10 min. All samples, including cell lysates, culture media, and collected dodecane phases, were again centrifuged at 13,000×*g* for 5 min, filtered using 0.2-μm-syringe filters, and analyzed using an Agilent HPLC system equipped with Kinetex 5 μm EVO C18 column (Phenomenex, Aschaffenburg, Germany) and Agilent UV detector at 203 nm. Metabolites were separated by isocratic elution with a flow rate of 1.0 mL/min at 30 °C for 30 min[18,38].

**Protein preparation for western blot analysis.** Overnight yeast cultures were inoculated into 50 mL YSC medium and grown at 30 °C, 250 rpm, for 2 days. Cell cultures were harvested to collect cells equivalent to an $OD_{600}$ of 100 for intracellular protein extraction, and supernatants were used to analyze extracellular proteins. Collected cell pellets were resuspended in 1 mL RIPA buffer (Thermo Scientific, MA, USA) with a 1x protease inhibitor cocktail (Roche Diagnostics, Mannheim, Germany), then the resuspensions were homogenized on ice using a Vibra-Cell VCX750 ultrasonic processor (SONICS, CT, USA) at 20% amplitude with 3 s intervals for 6 min. Extracellular proteins in supernatants were precipitated with 25% of ice-cold trichloroacetic acid (TCA) for 30 min, followed by centrifugation at 4000×*g* for 40 min at 4 °C. The precipitated protein pellets were washed with 1 mL ice-cold acetone three times and resuspended in 50 μl sterile water. The precipitated proteins were further concentrated using 30 kDa Amicon Ultra Centrifugal Filters (Millipore, Tullagreen, Ireland).

**Western blot analysis.** All protein samples were mixed with 5x SDS sample buffer (containing 250 mM Tris-HCl pH 6.8, 5% β-mercaptoethanol, 10% SDS, 0.5% bromophenol blue, and 50% glycerol), and then separated on 10% SDS-PAGE gels with Precision Plus Protein Standards (Bio-Rad, CA, USA). The gels were transferred onto PVDF membranes (Bio-Rad, CA, USA) and then blocked overnight with 4% (w/v) skim milk dissolved in TBST (Tris buffer saline with Tween-20). An anti-actin antibody (Chemicon International Inc., MA, USA, 1:5000) or an anti-His antibody (Santa Cruz Biotechnology, Inc., CA, USA, 1:1000) was used as a primary antibody. The primary antibodies were diluted in TBST for 1 h at room temperature. After washing with TBST three times, a peroxidase-conjugated Goat anti-Mouse IgG Antibody (Jackson Immun. Lab., USA, 1:5000) was used as a secondary antibody. The secondary antibody was incubated for 1 h at room temperature. After washing with TBST three times, the signals were visualized with Pierce ECL Western Blotting Substrate (Thermo Scientific, MA, USA) using the ChemiDoc MP imaging system (Bio-Rad, USA).

**Confocal fluorescence microscopy.** Yeast cells were cultured in the YSC medium supplemented with all amino acids without uracil to maintain plasmids expressing GFP-fused tSPF derivatives. After growing cells for 24 h at 30 °C with 250 rpm of shaking, cells were fixed for 20 min with 3.7% formaldehyde in PBS (pH 7.4) and observed using a multiphoton confocal microscope (Zeiss-LSM 780, Germany) equipped with a Plan-Apochromat 63 × oil immersion objective. Confocal images were processed and analyzed using ZEN imaging software 2.1 (Zeiss) and Image-Pro Plus 7.0.1 (Media Cybernetics).

**Identification of ligand binding sites of SPF.** The three-dimensional crystallography structure of SPF (PDB ID: 4OMK) was acquired from the RCSB protein data bank (pdb). For molecular docking using Discovery Studio 2020 v20.1.0.19, the A chain of 4OMK was used, in which the bound squalene and water molecules in the crystal structure were eliminated before simulation. In the crystallography structure, several amino acids (R4, K66, F113, C176, and Y202) are vibrating, but we intentionally chose one possible structure for each amino acid to construct a rigid protein structure. The active site for squalene binding of A chain was determined using Computer Atlas of Surface Topology of protein 3.0 (CASTp): identified residues for squalene binding were L84, I103, L106, A108, L111, L112, L120, L121, K124, I151, Y153, C155, L158, H162, A167, V168, A170, Y171, F174, L175, L186, L189, F198, A201, Y202, I205, L209, T213, and I217.

**Ligand selection.** Forty-three terpenes as sdf format were selected to molecular docking from PubChem compound database.; it includes 2,3-oxidosqualene (CID: 5366020), myrcen (CID: 5319723), (z)-ocimene (CID: 5320250), linalool (CID: 6549), genariol (CID: 637566), nerol (CID: 643820), limonene (CID: 22311), menthol (CID: 1254), α-pinene (CID: 6654), camphor (CID: 2537), carvacrol (CID: 10364), carvone (CID: 7439), β-farnesene (CID: 5281517), nerolidol (CID: 5284507), β-bisabolene (CID: 10104370), valencen (CID: 9855795), amorphadiene (CID: 11052747), capsidiol (CID: 161937), α-cedrane (CID: 9548702), artemisinin (CID: 68827), phytane (CID: 12523), phytol (CID: 5280435), camphorene (CID: 101750), retinol (CID: 445354), labdane (CID: 9548711), clerodane (CID: 182677), pimarane (CID: 9548698), abietane (CID: 6857485), tigliane (CID: 154992), kaurane (CID: 9548699), taxane (CID: 9548828), squalene (CID: 638072), lanosterol (CID: 246983), α-amyrin (CID: 73170), β-amyrin (CID: 73145), hopane (CID: 10115), oleanolic acid (CID: 10494), lycopene (CID: 446925), rubixanthin (CID: 5281252), β-carotene (CID: 5280489), lutein (CID: 5281243), zeaxanthin (CID: 5280899), and astaxanthin (CID: 5281224). Furthermore, hydrophilic ligands such as glutamate (CID: 33032) and pyruvate (CID: 107735) were added as control molecules. Downloaded sdf files were converted to pdbqt format by OpenBabel toolbox in PyRx 0.8 virtual tool.

**Molecular docking.** The molecular docking between SPF and potential ligands (43 terpenes) was implemented by AutoDock Vina 1.1.2 option of PyRx 0.8 virtual screening tool. Based on the aforementioned binding sites, the grid box for posing was assigned as 24.6222 × 29.3125 × 21.3921 Å, and the docking poses were scored with exhaustiveness 8. The A chain of 4OMK was considered a rigid body and docked with flexible terpene ligands. After the docking process, each ligand yielded several docking poses (up to nine poses) on the basis of root mean square deviation (RMSD) values. The first pose of each terpene, which has the highest score, were selected to compare the docked structures of ligands and binding affinities with SPF.

**Mass spectrometry.** Electrospray ionization-mass spectrometry (ESI-MS) experiment was performed by using Synapt G2-Si (Waters Corporation, MA, USA) mass spectrometer equipped with a nano-electrospray source. The sample solution of tSPF (~2 μM) and squalene (100 μM) prepared in water/methanol (1:1, v/v) was continuously transferred to the nano-electrospray emitter with a flow rate of 300 nL/min and nebulized with 2 kV electrospray voltage. The mass spectrum containing tSPF-squalene complex ions was obtained by averaging signals for 20 min.

**Figure preparation.** Figures were prepared using BioRender.Com for scientific illustrations.

**Statistics and reproducibility.** The western blot and confocal fluorescence microscopy experiments in Fig. 2D, E and in Supplementary Figs. 5, 6, 11A, B were performed at least triplicate at two independent times.

**Reporting Summary**. Further information on research design is available in the Nature Research Reporting Summary linked to this article.

## Data availability

All data in the main text and supplementary information are available from the corresponding authors upon reasonable request. The data for molecular docking are from the Protein Data Bank (PDB accession code: 4OMK) and the PubChem compound database under accession codes: 5366020, 5319723, 5320250, 6549, 637566, 643820, 22311, 1254, 6654, 2537, 10364, 7439, 5281517, 5284507, 10104370, 9855795, 11052747, 161937, 9548702, 68827, 12523, 5280435, 101750, 445354, 9548711, 182677, 9548698, 6857485, 154992, 9548699, 9548828, 638072, 246983, 73170, 73145, 10115, 10494, 446925, 5281252, 5280489, 5281243, 5280899, 5281224, 33032, and 107735. The processed data used to generate figures are provided in the Source Data file. Source data are provided with this paper.

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

## Acknowledgements

We thank Hyesung Jo and Jiyeon Lee for their assistance with mass spectroscopy measurements. This work was supported by the Basic Science Research Program (NRF-2021R1A2C2008074 and NRF-2017R1C1B3012050) through the National Research Foundation of Korea (NRF) grant funded by the Ministry of Science, ICT (MSIT). S.-H.S., J.-E.K., and J.Y.L. acknowledge funding from the Korea Research Institute of Chemical Technology through Core Program (SS2142-10).

## Author contributions

S.-H.S., J.-E.K., Y.-J.K., S.S.O., and J.Y.L. conceived the study and designed the experiments. G.P. performed the molecular docking simulation, J.S. conducted the mass spectroscopy experiment and B.H.S. contributed to the western blot analysis. All authors assisted in performing the experiments and data analysis. J.Y.L. supervised the research. S.S.O. and J.Y.L. wrote the manuscript with inputs from all other authors.

## Competing interests

The authors declare no competing interests.
