## [Peer Review File · Nature Communications]

Reviewers' comments:

Reviewer #2 (Remarks to the Author):

I appreciate the explanations given by the authors in their second rebuttal. I sustain that the findings of this study are interesting and potentially very valuable but that the weakness of this study lies in the poor mechanistic understanding of the observations. The choice of conditions to measure stoichiometry are now clarified – thank you. However, my recommendation to test microscopy, which could start to answer questions about the mechanism of export, was too quickly dismissed. Of course, I would not expect that one protein bound to 10 squalene molecules could be resolved by fluorescence microscopy. However, it is possible (one could argue even likely) that multiple protein-squalene complexes could cluster together to form protein-lipid droplets or rafts that may be resolvable via microscopy. The experiment is simple enough, and could begin to provide evidence for a mechanistic understanding of the system. Other experiments to begin to probe the mechanism of transport could be done instead; for example, test squalene secretion in strains with key gene deletions, but it seems to me that the microscopy suggestion is an easy and quick one to test first. Otherwise, the study will remain phenomenological in nature, which is still interesting and valuable, but not sure up to the standards typically found in this journal for this type of biotechnology papers; especially given that the authors concede they cannot claim that this export mechanisms can be used to selectively secrete terpenes in a programmable fashion, which was a big selling point and a claim of major impact in the original manuscript.

Reviewer #3 (Remarks to the Author):

The approach described in Son et al of using a secreted protein that can bind hydrophobic small molecules for secretion to the extracellular medium remains an intriguing if incompletely understood strategy. The authors' responses to previous reviews are appreciated, though much of the mechanistic descriptions remain out of reach. Given the novelty of the approach, but in the absence of further experimental data, I would thus recommend publication with minor revisions clarifying or eliminating claims that are still overselling the effectiveness of the strategy as it currently stands.

One major claim in the manuscript is that the product that is selectively secreted needs to accumulate in the ER where it will bind to tSPF (lines 106-119, Figs 1 and 2). While this proposed mechanism seems intuitive, it remains untested. For instance, what if the product is synthesized in another compartment all together? As all the hydrophobic products studied are surmised to be synthesized in the ER (squalene, lycopene, β -carotene) and no other compartments were investigated, the "series of coordinated actions" as described by the authors is at best a proposed model and not as firm as the current presentation suggests. Furthermore, given that no other carrier proteins or targeting tags are tested, the generality/versatility of this strategy for both other classes of metabolites as well as other subcellular locations is not yet proven, despite claims such as those made in lines 292-295. Without more data, claims like these should be taken out from the manuscript.

In addition, a persistent concern from previous reviews is that *S. cerevisiae* simply cannot meet the

demands secreting multiple g/L of carrier proteins for this strategy to be viable for most applications. The authors' responses to this have been rather vague and unspecific. While improvements are continuously being made to increase secretion of heterologous proteins in *S. cerevisiae*, I remain unconvinced that any of the listed strategies (Supp. Table 7, Supp. Note 5, etc.) would be sufficient to overcome that fundamental hurdle, especially in view of the burdens placed by overexpression of heterologous pathways necessary to produce the metabolites of interest in the first place. Instead, the discussion should not only acknowledge that this limitation exists, but also discuss how the strategy can still be useful in certain ways as it is, rather than stating and listing vague, as-yet-determined strategies that may or may not solve the hurdle eventually.

Point-by-Point Response

Title: Metabolite trafficking enables membrane-impermeable-terpene secretion by yeast

Corresponding Author: Dr. Ju Young Lee & Seung Soo Oh

Authors: So-Hee Son et al.

Reviewer #2

Reviewer comment:

I appreciate the explanations given by the authors in their second rebuttal. I sustain that the findings of this study are interesting and potentially very valuable but that the weakness of this study lies in the poor mechanistic understanding of the observations. The choice of conditions to measure stoichiometry are now clarified – thank you. However, my recommendation to test microscopy, which could start to answer questions about the mechanism of export, was too quickly dismissed. Of course, I would not expect that one protein bound to 10 squalene molecules could be resolved by fluorescence microscopy. However, it is possible (one could argue even likely) that multiple protein-squalene complexes could cluster together to form protein-lipid droplets or rafts that may be resolvable via microscopy. The experiment is simple enough, and could begin to provide evidence for a mechanistic understanding of the system. Other experiments to begin to probe the mechanism of transport could be done instead; for example, test squalene secretion in strains with key gene deletions, but it seems to me that the microscopy suggestion is an easy and quick one to test first. Otherwise, the study will remain phenomenological in nature, which is still interesting and valuable, but not sure up to the standards typically found in this journal for this type of biotechnology papers; especially given that the authors concede they cannot claim that this export mechanisms can be used to selectively secrete terpenes in a programmable fashion, which was a big selling point and a claim of major impact in the original manuscript.

Response: We greatly appreciate that the reviewer kept considering our work of terpene secretion to be “interesting and potentially very valuable.” Moreover, we are truly sorry that our previous response could not fully address your concerns. We tried to do our best again to improve our understanding of secretion mechanisms in meeting the high standard of *Nature*

Communications, and as the reviewer recommended, we further performed fluorescent microscopy experiments to find the evidence of multiple protein-squalene complex clusters as described below.

First, during squalene secretion by the engineered yeast strain (Suc2-tSPF-expressing SQ, Suc2-tSPF/SQ), we tried to observe colocalization of squalene and tSPF as a result of complex cluster formation using confocal fluorescence microscopy (**Supplementary Fig. 11A**). In the yeast cells, the green and red fluorescence by GFP-fused tSPF and Nile red-stained squalene overlapped each other, indicating the location of tSPF and squalene would be the same before secretion, presumably by formation of tSPF-squalene droplet clusters. Even in the extracellular milieu, the fluorescent overlapping was also observed, suggesting that even after secretion, tSPF and squalene would stay together, forming tSPF-squalene droplets.

Second, in terms of squalene-tSPF complexes with an uneven stoichiometry, we intentionally prepared squalene droplets (100 ~ 400 nm in diameter) and mixed with GFP-fused tSPFs of which concentration (~0.1 mg/mL) was 100 times less than that of squalene (~10 mg/mL). As shown in **Supplementary Fig. 11B**, the locations of squalene droplets and tSPFs were observed to be the same by the confocal fluorescence microscopy; this result indicates that the tSPF protein can bind not only to the single molecule of squalene but also to the squalene cluster or droplet. We note that without tSPF fusion, the GFP proteins did not bind to the squalene droplets.

Although there would remain a gap in fully understanding the mechanism of tSPF-driven terpene secretion, our previous electrospray ionization-mass spectrum (**Supplementary Fig. 10**) and current fluorescence microscopy observation (**Supplementary Fig. 11**) are considered to support our claim, “tSPF is supposed to form squalene clusters and carry multiple squalene molecules due to a driving force of hydrophobic interactions.” To provide clues in further understanding the mechanism of the carrier protein-driven terpene secretion, we added **Supplementary Fig. 11** and further discussions in the revised Supplementary Information (see the updated **Supplementary Note 4**).

New Supplementary Fig. 11.

Supplementary fig. 11 *In vivo* and *in vitro* evaluation of cluster formation by multiple tSPF-squalene complexes using confocal fluorescence microscopy.

(A) When GFP-fused Suc2-tSPFs were overexpressed by squalene-producing SQ strains, the green fluorescence by the GFP-fused tSPFs and the red one by Nile red-stained squalene were colocalized inside and outside the cells (left and right, respectively), presumably by formation of tSPF-squalene droplet clusters. **(B)** Squalene droplets (100 ~ 400 nm in diameter) were prepared (left) and subsequently mixed with GFP proteins (middle) and GFP-fused tSPFs (right) of which concentrations (~0.1 mg/mL) were 100 times less than that of squalene (~10 mg/mL). Unlike the GFPs only, the GFP-fused tSPFs bound to the squalene droplets as

evidenced by their identical locations; from this observation, it is supposed that the tSPF could bind not only to the single squalene molecule but also to its forms of clusters or droplets. For the confocal fluorescence microscopy observation, squalene was stained with Nile red (excitation at 543 nm and emission at 585 nm), but GFP was fluorescent (excitation at 488 nm and emission at 507 nm). Confocal fluorescence microscopy images were processed and analyzed using ZEN imaging software (Zeiss). Scale bar: 5 μm .

Reviewer #3

Reviewer comment:

The approach described in Son et al of using a secreted protein that can bind hydrophobic small molecules for secretion to the extracellular medium remains an intriguing if incompletely understood strategy. The authors' responses to previous reviews are appreciated, though much of the mechanistic descriptions remain out of reach. Given the novelty of the approach, but in the absence of further experimental data, I would thus recommend publication with minor revisions clarifying or eliminating claims that are still overselling the effectiveness of the strategy as it currently stands.

Overall response: We sincerely thank that the reviewer understood the novelty of our work about tSPF-driven-terpene secretion and kept considering it to be “intriguing.” Moreover, we are truly sorry that our previous description of secretion mechanism was not fully addressed. To strengthen our understanding in the carrier protein-driven terpene secretion, we additionally performed confocal fluorescence microscopy experiments to confirm formation of tSPF-squalene droplet complexes during their cellular secretion (**Supplementary Fig. 11A**). Briefly, we successfully observed colocalization of Nile red-stained squalene droplets with GFP-fused tSPFs both inside and outside squalene-producing yeast cells, which can be explained by the formation of tSPF-squalene droplet clusters. In terms of squalene-tSPF complexes with an uneven stoichiometry, we intentionally prepared squalene droplets (100 ~ 400 nm in diameter) and mixed with GFP-fused tSPFs of which concentration (~0.1 mg/mL) was 100 times less than that of squalene (~10 mg/mL) (**Supplementary Fig. 11B**). From the observation that squalene droplets and tSPFs were colocalized, it is supposed that the tSPF protein could bind not only to the single squalene molecule but also to its forms of clusters or droplets. Our previous electrospray ionization-mass spectrum (**Supplementary Fig. 10**) and current fluorescence microscopy observation (**Supplementary Fig. 11**) would be supportive for our claim, “tSPF is supposed to form squalene clusters and carry multiple squalene molecules due to a driving force of hydrophobic interactions.” For better understanding the terpene secretion mechanism, we added **Supplementary Fig. 11** and further discussions in the revised Supplementary Information (see the updated **Supplementary Note 4**).

Additionally, as the reviewer pointed out, we have toned down or eliminated all overselling statements of the effectiveness of our strategy in the revised manuscript to avoid misleading. The changes in the revised manuscript were highlighted in red accordingly.

Reviewer comment 1:

One major claim in the manuscript is that the product that is selectively secreted needs to accumulate in the ER where it will bind to tSPF (lines 106-119, Figs 1 and 2). While this proposed mechanism seems intuitive, it remains untested. For instance, what if the product is synthesized in another compartment all together? As all the hydrophobic products studied are surmised to be synthesized in the ER (squalene, lycopene, β -carotene) and no other compartments were investigated, the “series of coordinated actions” as described by the authors is at best a proposed model and not as firm as the current presentation suggests. Furthermore, given that no other carrier proteins or targeting tags are tested, the generality/versatility of this strategy for both other classes of metabolites as well as other subcellular locations is not yet proven, despite claims such as those made in lines 292-295. Without more data, claims like these should be taken out from the manuscript.

Response: We agree with the reviewer that we only explored the secretion of the hydrophobic products that are synthesized in the ER. As prior studies succeeded in signal peptide-guided protein secretion [1], we suggested that other types of carrier proteins would be secreted together with their target metabolites (lines 292-295 of the original manuscript); however, we also admit that we did not prove the generality and versatility of our secretion strategy by introduction of different carrier proteins or signal peptides for other metabolites or subcellular locations. Accordingly, we eliminated the possibility description about other metabolite delivery to desired intracellular locations, and as described below, the sentence was amended to limitedly describe the potential expansion for our carrier protein-based secretion.

Amended text: “(lines 292-295) **Furthermore, use of other carrier proteins and signal peptides could be explored for our metabolite trafficking strategy with potential applications for metabolic engineering and synthetic biology.**”

[1] Hou J *et al.* Metabolic engineering of recombinant protein secretion by *Saccharomyces cerevisiae*. 2012. *FEMS Yeast Res.* 12, 491-510

Reviewer comment 2:

In addition, a persistent concern from previous reviews is that *S. cerevisiae* simply cannot meet the demands secreting multiple g/L of carrier proteins for this strategy to be viable for most applications. The authors’ responses to this have been rather vague and unspecific.

While improvements are continuously being made to increase secretion of heterologous proteins in *S. cerevisiae*, I remain unconvinced that any of the listed strategies (Supp. Table 7, Supp. Note 5, etc.) would be sufficient to overcome that fundamental hurdle, especially in view of the burdens placed by overexpression of heterologous pathways necessary to produce the metabolites of interest in the first place. Instead, the discussion should not only acknowledge that this limitation exists, but also discuss how the strategy can still be useful in certain ways as it is, rather than stating and listing vague, as-yet-determined strategies that may or may not solve the hurdle eventually.

Response: We apologize that we could not sufficiently acknowledge the possible limitations of our carrier protein-based terpene secretion strategy; in particular, the overexpression limit of heterologous proteins that are necessary to produce metabolites of interest, as well as carrier proteins, should have been well-described. However, as an initial proof-of-concept study, our work suggests a conceptual methodology to secrete highly valuable, yet membrane-impermeable products to extracellular spaces, and even with the incomplete terpene secretion, our terpene trafficking strategy holds promises for further scientific studies and eventual applications (*e.g.*, regulation of overflow metabolism and semi-continuous fermentation). As the reviewer requested, we clearly described the possible limitations and advantages of our strategy in the revised manuscript.

(lines 272-287) For further improvement of terpene production and secretion, increasing the levels in overexpression of heterologous proteins relevant to terpene and carrier protein synthesis would be inevitably necessary, which is critical and not solved for our metabolite trafficking strategy. Moreover, as the underlying mechanisms of our carrier protein-driven terpene secretion has not been fully understood yet, the efforts to unravel the mechanisms would be also important. For the eventual applications, many different ways would be devised by systematically combining various synthetic biology tools and strategies (Supplementary Table 7 and Supplementary Note 5). For this proof-of-concept study...”

REVIEWERS' COMMENTS

Reviewer #2 (Remarks to the Author):

I am glad the new microscopy experiments begin to suggest a mechanism of action. I am satisfied with this version of the manuscript and agree with its publication.

Reviewer #3 (Remarks to the Author):

I thank the authors for the added microscopy experiment, I believe the manuscript is improved by its inclusion and now merits publication.